# Autophagy Dysregulation in Metabolic Associated Fatty Liver Disease: A New Therapeutic Target

**DOI:** 10.3390/ijms231710055

**Published:** 2022-09-02

**Authors:** Chun-Liang Chen, Yu-Cheng Lin

**Affiliations:** 1Department of Pediatrics, Far Eastern Memorial Hospital, New Taipei City 220, Taiwan; 2School of Medicine, National Yang Ming Chiao Tung University, Taipei City 112, Taiwan

**Keywords:** autophagy, MAFLD, fatty liver disease, metabolic disease, NAFLD

## Abstract

Metabolic associated fatty liver disease (MAFLD) is one of the most common causes of chronic liver disease worldwide. To date, there is no FDA-approved treatment, so there is an urgent need to determine its pathophysiology and underlying molecular mechanisms. Autophagy is a lysosomal degradation pathway that removes damaged organelles and misfolded proteins after cell injury through endoplasmic reticulum stress or starvation, which inhibits apoptosis and promotes cell survival. Recent studies have shown that autophagy plays an important role in removing lipid droplets from hepatocytes. Autophagy has also been reported to inhibit the production of pro-inflammatory cytokines and provide energy for the hepatic stellate cells activation during liver fibrosis. Thyroid hormone, irisin, melatonin, hydrogen sulfide, sulforaphane, DA-1241, vacuole membrane protein 1, nuclear factor erythroid 2-related factor 2, sodium-glucose co-transporter type-2 inhibitors, immunity-related GTPase M, and autophagy-related gene 7 have been reported to ameliorate MAFLD via autophagic induction. Lipid receptor CD36, SARS-CoV-2 Spike protein and leucine aminopeptidase 3 play a negative role in the autophagic function. This review summarizes recent advances in the role of autophagy in MAFLD. Autophagy modulates major pathological changes, including hepatic lipid metabolism, inflammation, and fibrosis, suggesting the potential of modulating autophagy for the treatment of MAFLD.

## 1. Introduction

Metabolic associated fatty liver disease (MAFLD) is one of the most common causes of chronic liver disease worldwide. The global prevalence of MAFLD is thought to be constantly increasing and is currently estimated to be ~25% [1]. MAFLD was previously known as non-alcoholic fatty liver disease (NAFLD) [2,3], defined as the excessive deposition of fat in the liver in the absence of heavy alcohol consumption. Since fatty liver associated with metabolic dysfunction is common, “MAFLD” has recently been recognized as a more appropriate overarching term [3].

To date, there is no FDA-approved treatment, so there is an urgent need to determine its pathophysiology and underlying molecular mechanisms [4]. The pathogenesis of MAFLD is multifactorial and is strongly associated with obesity and insulin resistance [5,6]. Liver with steatosis is more vulnerable to secondary injuries. This is popularly known as the two-hit hypothesis [7]. The first hit is due to hepatic fat overload, which is associated with obesity and insulin resistance. The second hit is mediated by oxidative stress and lipid peroxidation and inflammatory cytokines which cause inflammation and fibrosis.

A fatty liver condition develops when fat content accounts for more than 5% of the liver [8]. During the development of MAFLD, multicellular interactions occur involving hepatocytes, hepatic stellate cells (HSCs), hepatic macrophages (also called Kupffer cells, KCs), and adipose tissue [9,10]. Fat accumulation leads to lipotoxicity in hepatocytes, which secrete exosomes to activate HSCs [11]. A high fat diet (HFD) causes gut microbiota dysbiosis and results in the transfer of bacterial endotoxin to the liver through the gut-liver axis [12] and activates KCs to produce proinflammatory cytokines (IL-6, IL-1β and TNF-α) and profibrotic mediators (TGF-β) [13] to activate HSCs.

Autophagy plays an important role in removing lipid droplets from hepatocytes [14]. In addition, non-parenchymal cells of the liver, including KCs and HSCs also can utilize autophagy to maintain their homeostasis or function, thereby affecting pro-inflammatory and fibrotic responses in MAFLD progression [15].

This review summarizes the potential interactions between these cells and tissues, focusing on recent findings on the impact of regulatory mechanisms on hepatic autophagy. The effects of autophagy-modulating drugs and proteins on hepatic lipid metabolism may be a new therapeutic target for MAFLD.

## 2. Hepatic Parenchymal and Non-Parenchymal Cells and Adipose Tissue in MAFLD

In addition to hepatocytes, non-parenchymal cells (including KCs and HSCs) and adipose tissue are closely related to the pathogenesis of MAFLD [9,10].

### 2.1. Hepatocyte in MAFLD

If a subject has a long-term high carbohydrate- and/or high fat-containing diet, excess lipid droplets accumulate in hepatocytes and result in a fatty liver, causing endoplasmic reticulum (ER) stress and lipotoxicity [14,16]. Lipotoxic hepatocytes secrete exosomes containing microRNA (miRNA), mRNA, long non-coding RNA (lncRNAs), DNA, and proteins [17], which in turn accelerate the activation of HSCs and lead to fibrosis [18]. Luo, X. et al. purified miR-1297 from exosomes secreted by palmitic acid induced lipotoxic hepatocyte and demonstrated that miR-1297 promoted the activation of HSCs by inhibiting the PTEN/PI3K/AKT signaling pathway [11].

In normal liver tissue, the oxidative and antioxidant systems are in a state of homeostasis. Physiologically reactive oxygen species (ROS) are removed from hepatocytes by intracellular antioxidant mechanisms. However, under NAFLD pathological conditions, more ROS are generated with impaired ROS scavenging mechanisms, leading to oxidative stress and mitochondrial dysfunction. As a result, hepatocyte damage and apoptosis occur [19]. Li, J. et al. recently reported that hesperetin decreased hepatic oxidative stress and inflammation through a PI3K/AKT-Nrf2-ARE-dependent signaling pathway in cultured hepatocytes and in an HFD-induced rat model of NAFLD [19].

### 2.2. Macrophage in MAFLD

Macrophage is an essential element of innate immunity and play a central role in inflammation and host defense. Hepatic macrophages contain predominantly resident KCs that originate from the yolk sac and infiltrated bone marrow-derived monocytes from the circulation [10]. Macrophage is the master regulator of liver inflammation and fibrosis, producing proinflammatory cytokines (IL-6, IL-1β, and TNF-α) and profibrotic mediators (TGF-β and platelet derived growth factor, PDGF) after stimulation [13]. Previous studies have demonstrated that HFD altered the composition of the gut microbiota and bacterial metabolites in humans and mice [20]. The activity of KCs is affected by intestinal bacteria and endotoxins (lipopolysaccharide, LPS) that are transferred to the liver through the gut-liver axis [12]. Growing evidence points to the important role of macrophages in the development of NAFLD [21].

Signals from the microenvironment, including cytokines, growth factors, pathogen-derived molecules and fatty acids, cause macrophages to differentiate into sub-phenotypes with specific biological functions [22]. Macrophage can undergo classical M1 activation (stimulated by TLR ligands and IFN-γ) or M2 activation (stimulated by IL-4 or IL-13).

The M1 phenotype is characterized by the expression of high levels of proinflammatory cytokines, large amounts of reactive nitrogen and oxygen intermediates, the promotion of the Th1 response and strong microbicidal and tumoricidal activity. M2 macrophages are involved in the containment of parasites, the promotion of tissue remodeling and tumor progression and have an immunoregulatory function [23].

Wan, J. et al. demonstrated that an increase in M2 KCs promoted the apoptosis of M1 KCs, protecting hepatocytes from injury and the development of alcoholic fatty liver disease (ALD) and NAFLD [24]. Bleriot, C. et al. used a highly dimensional approach for a mouse model to demonstrate two distinct populations (KC1 and KC2) of KCs, which express different genes and proteins. The minor KC2 population expresses genes involved in metabolic processes, including fatty acid metabolism related to obesity and hepatic steatosis [25].

### 2.3. Hepatic Stellate Cells (HSCs) in MAFLD

HSCs are the major collagen-producing cells in the liver and play a key role in the progression of MAFLD [26]. TGF-β1 promotes fibrosis by inducing the differentiation of HSCs from the quiescent state into activated myofibroblasts, which increases the proliferation, migration, and expression of tissue inhibitors of matrix metalloproteases (TIMPs). These processes inhibit the degradation of the extracellular matrix (ECM) proteins and directly promote the synthesis of interstitial fibrillar collagens [27].

Previous studies demonstrate that glycosylation-dependent galectin-1/neuropilin-1 interactions promote liver fibrosis via the activation of TGF-β- and PDGF-like signals in HSCs [28]. A recent study showed that TGF-β-stimulated clone 22D4 (TSC22D4), a member of the TSC-22 family proteins, contributed to the TGF-β1-mediated activation of HSCs and promoted their proliferation and migration [29].

Regulation of autophagy in HSCs is complex and is associated with their activation [30,31]. Recently, Lucantoni, F. et al. showed that autophagy plays a dual role in HSC activation to prevent production of pro-fibrotic mediators or lead to cell death of HSCs [32].

### 2.4. Adipose Tissue in MAFLD

Adipose tissue contributes to the development of MAFLD by increasing the flux of free fatty acids (FFAs) to the liver and via the secretion of cytokines and adipokines (such as adiponectin, leptin, FABP4, FGF21, VEGF, IL-6, IL-8, and TNF-α) [33]. Different adipokines have different functions, for example, leptin stimulates the activation of KCs and HSCs, while adiponectin inhibits this process [33]. Tanaka, M. et al. recently demonstrated that circulating level of FABP4 is an effective marker to predict MAFLD in middle-aged and elderly human [34].

Visceral adipose tissue (VAT) and subcutaneous adipose tissue (SCAT) have different metabolic functions. Secretory products increased only in VAT but not SCAT are associated with metabolic abnormalities [33]. Excessive lipolysis in VAT leads to increased flow of FFA to the liver through the portal vein [35,36]. Previous studies showed that exercise stimulated the release of muscle-derived myokines from skeletal muscle, which in turn activated lipophagy and promoted browning of white adipocytes and ameliorated lipid accumulation in the liver.

The accumulation of macrophages in adipose tissue causes increased secretion of pro-inflammatory cytokines, such as TNF-α and IL-6. This is a common observation in patients with obesity and insulin resistance. Anti-inflammatory drugs, such as vitamin E and anti-TNF-α antibodies, reduced the levels of pro-inflammatory cytokines and improved insulin resistance and fatty liver disease [37].

## 3. Autophagy in MAFLD

Autophagy is an intracellular process involving the lysosomal degradation pathway that removes damaged organelles and misfolded proteins following cellular damages caused by ER stress or starvation. Autophagy inhibits apoptosis and promotes cell survival. The dysregulation of autophagy is associated with the development of various human diseases such as metabolic syndrome, pulmonary disorders, renal diseases, infectious diseases, cardiovascular disease, hepatic disorders, neurodegenerative disorders, and cancers [38,39].

The process for autophagy has three stages: initiation, maturation/nucleation, and degradation/expansion [39,40]. Various autophagy-related genes (ATG) proteins mediate this process. Previous studies demonstrated that there are at least 41 ATGs in yeast cells, but their homologues in mammalian cells are not fully understood [41]. The regulation of autophagy contributes to the formation of microtubule-associated protein 1 light chain-3 (LC3) and Beclin-1 [42]. Beclin-1 and the anti-apoptotic Bcl-2 family interact with each other to regulate autophagy. The Bcl-2/Beclin-1 complex also plays a key role in autophagy and apoptosis crosstalk [43].

Recent studies have shown that autophagy plays an important role in lipid removal in hepatocytes [14]. Autophagy has also been reported to inhibit the production of pro-inflammatory cytokines, inhibit adipose tissue browning [44], and to provide energy for the activation of HSCs during liver fibrosis (Figure 1). Next, we discuss newly discovered proteins/compounds that affect MAFLD phenotype by modulating autophagy.

### 3.1. Autophagic Inducer in MAFLD

Several compounds, proteins or genes induce autophagy, including thyroid hormone, irisin (an exercise-derived myokine), hydrogen sulfide, melatonin, DA-1241, vacuole membrane protein 1 (VMP1), nuclear factor erythroid 2-related factor 2 (Nrf2), sodium-glucose co-transporter type-2 inhibitors (SGLT-2i), immunity-related GTPase M (IRGM) and autophagy-related gene 7 (*ATG7*) (Table 1).

#### 3.1.1. Thyroid Hormone

Thyroid hormone (TH) maintains metabolic homeostasis at systemic and hepatic levels, which affects the de novo lipogenesis, cholesterol (CHO) metabolism, β-oxidation, and carbohydrate metabolism [45,46]. Growing evidence links TH levels to the risk of MAFLD [47]. TH regulates the metabolism of fatty acids and the disruption of TH function in the liver contributes to the development of MAFLD.

Previous studies reported that cellular TH signaling is altered in several liver-related diseases, including MAFLD and hepatocellular carcinoma (HCC) [48]. The agonists of TH and thyroid hormone receptor β (TRβ) can decrease hepatic TG content [49]. Recent clinical trials showed that hepatic steatosis was improved in patients with NAFLD after treatment with low-dose levothyroxine or TRβ-selective agonist, resmetirom (MGL-3196) [50].

Zhou, J. et al. demonstrated that TH decreased inflammation, hepatic steatosis, and fibrosis in TRβ over-expressed human hepatic cell line and in a mouse NASH model induced by a western diet supplemented with 15% fructose in drinking water [51]. After TH treatment, fatty acid oxidation was increased and lipotoxicity was decreased in cultured hepatic cells. TH treatment decreased hepatic steatosis, inflammasome and oxidative stress activation, production of chemokines and inflammatory cytokines, and collagen formation in the liver of mice with NASH. The effects of metabolic pathways related to hepatic organic acid, CHO, carbohydrate and fatty acid metabolism were increased. In addition, TH treatment increased hepatic autophagy and promoted β-oxidation of fatty acids [51].

#### 3.1.2. Irisin (an Exercise-Derived Myokine from Contractile Muscle)

Irisin activates hepatic autophagy, reduces liver inflammation, promotes the browning of white adipocytes, reverses intestinal epithelial barrier dysfunction, induces mitochondrial function, and improves the NAFLD phenotype [52,53,54,55,56,57,58,59,60,61].

The adaptor protein, myeloid differentiation factor 2 (MD2), interacts with Toll-like receptor 4 (TLR4) [62]. Bacterial LPS is recognized by the MD2-TLR4 complex, causes the recruitment of intracellular adaptor protein myeloid differentiation factor 88 (MyD88), and activates the downstream factors, mitogen-activated protein kinase (MAPK), and nuclear factor-κB (NF-κB), resulting in the release of pro-inflammatory cytokines [63]. Metabolism-associated molecular patterns (MAMPs), including glucose, FFAs, and CHO (all of which are present in excess in chronic metabolic diseases) activate the MD2-TLR4 signaling pathway [64].

Zhu, W. et al. compared liver histology, serum aspartate aminotransferase (AST), and alanine aminotransferase (ALT) levels, the levels of pro-inflammatory cytokines, and liver steatosis/fibrosis markers in a normal chow diet, HFD, and HFD in male C57BL/6 mice with and without exercise treatment. HFD significantly increased body weight, liver weight, AST and ALT levels, macrophage infiltration levels in liver, pro-inflammatory cytokines (IL-6, IL-1β, and TNF-α), the NAFLD activity score, liver steatosis, lobular inflammation, hepatocyte ballooning, liver collagen area, liver TGF-β1, and type I collagen expression levels. In contrast, these negative effects of HFD were significantly reduced after 6 weeks of exercise therapy [55].

Furthermore, the irisin content in soleus, gastrocnemius, and quadriceps muscles and plasma were significantly increased after exercise. Irisin counteracted the palmitic acid (PA) effects on the expression levels of TGF-β1 and type I collagen in murine hepatocyte AML12 cells, and decreased proinflammatory cytokine production in murine macrophage RAW264.7 cells. Mechanistically, irisin directly binds with MD2 to compete the binding of MD2 with TLR4 to improve NAFLD in mouse models [55].

Aged liver from elderly donors has decreased repair capacity following ischemia-reperfusion (IR) injury after liver transplantation [65]. Poor prognosis after hepatic IR in elderly patients may be due to weak hepatocyte autophagy and poor mitochondrial function [66]. Bi, J. et al. compared the autophagy activity between 3-month (young)- and 22-month (old)- old male Sprague–Dawley rats and found that the serum and liver levels of irisin in aged rats were decreased, and the expression of the autophagy marker LC3B was decreased. Apoptotic cells, the liver necrosis area, and p62 expression were also increased in aged rats. These adverse effects were then restored by supplementing with irisin [60].

#### 3.1.3. Hydrogen Sulfide

After carbon monoxide (CO) and nitrous oxide (NO), hydrogen sulfide (H_2_S) is the third gaseous signal molecule associated with smooth muscle relaxation in the vascular system [67]. H_2_S has other important biological functions and is involved in various human diseases, including MAFLD [68].

Wu, D. et al. used oleic acid (OA)-treated liver cells and HFD-fed C57BL/6J mice and found that OA-treatment decreased endogenous H_2_S levels, lipid level, cellular content of total cholesterol (TC) and triglyceride (TG), and the proliferation of liver cells, but increased apoptosis. In contrast, NaHS (an H_2_S donor) decreased the lipid accumulation and the apoptosis in OA-treated liver cells and increased the expression levels of Beclin-1 and the LC3 II/I ratio [69]. Increased body weight, liver weight, white fat and brown fat weight, liver TC and TG content, AST, ALT, TNF-α, IL-1β, and IL-6 levels in HFD-fed mice were significantly attenuated by H_2_S. The impaired hepatic autophagic flux in HFD-fed mice was restored by H_2_S. H_2_S inhibited the ROS/PI3K/AKT/mTOR signaling pathway in the liver [69].

Guo, J. M. et al. used LPS-treated chicken and also found that H_2_S decreased the ROS/PI3K/AKT/mTOR signaling pathway and significantly increased the expression levels of autophagic markers (Beclin1, ATG5, and LC3 II/I ratio) [70].

#### 3.1.4. Melatonin

Melatonin (5 methoxy-N-acetyltryptamine) is primarily produced by the pineal gland at night and circulates around the entire body [71]. Dark environment increases its synthesis and secretion, which is inhibited by light [72]. Melatonin is distributed in the skin, the gastrointestinal tract, bone marrow, and the liver. The largest accumulation is in the liver [73]. Melatonin regulates various signaling pathways related to oxidative stress, apoptosis, cell damage, and inflammation [74,75,76]. Several studies suggest melatonin may play a role in the development of liver fibrosis, hepatotoxicity, and MAFLD [77,78].

San-Miguel, B. et al. collated studies over the past 10 years on the anti-fibrotic effects of melatonin and discuss the cellular and molecular mechanisms that prevent progression of liver fibrosis, cirrhosis, and liver cancer [79]. Two studies using human HSCs cells LX-2 and the primary isolated HSCs showed that the expression levels for liver fibrosis markers (α-SMA, type 1 collagen and TGF-β) decreased after melatonin treatment [80,81]. Several animal studies showed that melatonin decreased the expression levels for liver fibrosis markers TGF-β, MMP9, and TIMP1 [79]. Of the three studies discussing the regulation of autophagy by melatonin, two showed a positive effect and one showed a negative effect on autophagy activity [79].

Barangi, S. et al. studied the effects of melatonin on Razi mice treated with BaP, benzo(a)pyrene, an environmental pollutant. They found that BaP treatment increased serum AST and ALT levels, liver malondialdehyde level, liver histopathology scores, and cleaved caspase-3, and decreased LC3 II/I ratio and Beclin-1 expression levels. Concomitant administration of melatonin and BaP significantly suppressed the effect of BaP alone [82].

#### 3.1.5. DA-1241 (a Novel GPR119 Agonist)

GPR119 is a G protein-coupled receptor (GPCR) that is primarily expressed in the pancreas and gastrointestinal tract. When nutrients are ingested, GPR119 is activated in pancreatic beta cells, and then Gs-coupled adenylyl cyclase is activated to increase intracellular cyclic adenosine monophosphate (cAMP) and lead to glucose-dependent secretion of insulin.

Kim, M. K. et al. isolated the GPR119 agonist, DA-1241, from more than 2000 in-house synthesized compounds and showed better intrinsic efficacy than previous clinical candidates in vitro and in vivo. This treatment is currently in early clinical development for the treatment of T2D [83].

GPR119 is also expressed in the liver. Kim, Y. et al. treated HFD-fed C57BL/6J mice with DA-1241 after the onset of hyperglycemia and found that DA-1241 increased autophagic flux, as measured by mRFP-GFP-LC3 signaling. DA-1241 also caused significant reductions in serum CHO and hepatic TG in HFD mice [84].

#### 3.1.6. ER Transmembrane Protein-Vacuole Membrane Protein 1 (VMP1)

VMP1 is an ER transmembrane protein that regulates the formation of autophagosomes and lipid droplets [85,86]. Morishita, H. et al. reported that *vmp1* gene deficiency resulted in lipoprotein accumulation in the liver and gut of zebrafish. It caused lipid accumulation in the visceral endoderm and gut in *Vmp1*-deficient mice [87].

Recently, Jiang, X. et al. used hepatocyte-specific *Vmp1* knockout mice to demonstrate that the deletion of *Vmp1* impaired mitochondrial β-oxidation and caused the accumulation of neutral lipids in the liver. This study measured lipoprotein secretion and found that hepatocyte-specific *Vmp1* knockout mice had significantly reduced serum TG levels and TG secretion rates. Decreased VMP1 expression levels have been reported in human NASH liver samples and in the liver of a mouse model of HFD-induced NASH [88].

#### 3.1.7. Nuclear Factor Erythroid 2-Related Factor 2 (Nrf2)

Ischemia results in oxidative stress and damage to cells, tissues, and organs. Reperfusion through revascularization and restoration of blood flow after ischemia causes further damage [89]. During oxidative stress, Nrf2 is a master regulator of homeostasis of redox. Upon activation, Nrf2 translocates into the nucleus and upregulates the expression levels of downstream factors, including HO-1, GPx, NQO1, and SOD, thereby inhibiting ROS production and protecting cells from apoptosis [90].

Sadrkhanloo, M. et al. reviewed the roles of Nrf2 in the alleviation of ischemia-reperfusion in the liver, brain, lung, testis, and heart [90]. Ischemia/reperfusion (I/R) injury often occurs after liver transplantation. Mitophagy (a special type of autophagy that degrades damaged mitochondria [91]) protects hepatic cells from reperfusion injury. Activation of the Nrf2/HO-1 axis has been reported to reduce inflammation and oxidative stress and induce autophagy in mouse and rat models of I/R injury [92,93,94].

Using an I/R injury mouse model and *Nrf2* knockout mice, Xu, D. et al. showed that pretreatment with Nrf2 activator significantly reduced ALT and AST levels, induced Nrf2/HO-1-dependent pathways to inhibit apoptosis markers, cleaved caspase-3 and Bcl-2, and induced the expression of the autophagy marker LC3B in a mouse model of I/R injury [92].

#### 3.1.8. Sodium-Glucose Co-Transporter Type-2 Inhibitors (SGLT-2i)

SGLT-2 is a cell membrane exchange transporter that is located in the proximal tubule and promotes the reabsorption of glucose by the kidneys. The first SGLT-2i was isolated from an apple tree and used to treat infectious diseases in 1838 [95]. Till now, a number of SGLT-2i have been discovered and used to control blood sugar, body weight, and serum uric acid. Regulatory effects of SGLT-2i have been implicated in ER and oxidative stress, autophagy, apoptosis, and inflammation [96]. Ye, T. et al. treated HFD-fed C57BL/6J mice with empagliflozin and found that empagliflozin reduced fat mass, body weight, plasma TG and FFA levels, fasting glucose levels, and NLRP3 inflammasome activity, and induced HO-1-adiponectin-dependent signaling pathway [97].

Androutsakos, T. et al. recently summarized the current evidence for the effects of SGLT-2i on MAFLD in cellular and animal models as well as in human trials [96]. Most SGLT-2i inhibited cell proliferation by activating caspase 3 in human hepatocellular carcinoma cells—HepG2 cells [98]. Li, L. et al. demonstrated that dapagliflozin, an SGLT-2i, reduced lipid droplet content in HepG2 and L02 (an immortalized normal human hepatocyte-derived hepatocyte) cells. Dapagliflozin induces autophagy via the AMPK-mTOR pathway in ZDF rats. In addition, dapagliflozin reduced lipid accumulation in the liver of ZDF rats by decreasing lipogenic enzymes, while inducing fatty acid oxidation enzymes [99].

In human, a clinical trial conducted from 2017 to 2021 enrolled 25 patients with T2DM and NAFLD and found that SGLT-2i treatment resulted in significant improvements in plasma liver enzymes, liver steatosis, and NAFLD fibrosis scores [96].

#### 3.1.9. Immunity-Related GTPase M (IRGM)

IRGM controls the core autophagy machinery by interacting with NOD2 (nucleotide-binding oligomerization domain protein 2) and ATG16L1 [40]. IRGM stabilizes the Thr172 phosphorylated form of adenosine monophosphate-activated protein kinase (AMPK) [100], which is required for AMPK activation and the further phosphorylation of the ULK1 complex and Beclin-1 to induce autophagy [101,102]. IRGM are functionally relevant to inflammatory and autophagic responses [40]. Genetic susceptibility is a well-known factor for the risk of NAFLD [103,104]. Genome-wide association studies have shown that the autophagy gene *IRGM* is a risk allele for Crohn’s disease [105].

We have previously shown that genetic variants in *IRGM* alter hepatic lipid metabolism through lipophagy and contribute to the risk of NAFLD in obese children and adolescents [106]. We enrolled 832 obese children and adolescents, 22.8% of whom were NAFLD patients. We found that *IRGM* rs10065172 variant significantly increased the risk of MAFLD. *IRGM* knockdown in human hepatocellular carcinoma cell lines (HepG2 and PLC/PRF/5 cells) reduced the expression levels of autophagy markers pULK1, Beclin-1, and the LC3 II/I ratio, resulting in the accumulation of lipid droplets in hepatocytes. In contrast, overexpression of *IRGM* or treatment with the autophagy inducer rapamycin induced autophagy and reduced lipid droplet accumulation in hepatocytes [106].

#### 3.1.10. Autophagy-Related Gene 7 (*ATG7*)

Baselli, G. A. et al. performed whole-exome sequencing for 301 patients with NAFLD and showed enriched variants of *ATG7* rs143545741 p.V471A and p.P426L (loss-of-function mutation) in patients with advanced fibrosis [107]. *ATG7* knockdown in primary human hepatocytes cultured in 2D and 3D increased the intra-cellular lipid droplets. Overexpression of V471A and P426L mutants in HepaRG cells, an immortalized hepatic cell line derived from primary human hepatocytes, resulted in significantly reduced lipid droplets. V471A mutation significantly increased p62 expression and decreased the LC3 II/I ratio in HepG2 cells [107].

Kim, K.H. et al. showed that skeletal muscle-specific *Atg7* knockout mice exhibited reduced lipid accumulation and increased β-oxidation-related genes. *Atg7* knockout mice fed with HFD had lower expression of lipogenic genes in the liver, protecting against diet-induced obesity and insulin resistance. Metabolic outcomes after initiation of autophagy or lipophagy in muscle differ from those in liver [108,109]. A myokine microarrays reveal that fibroblast growth factor 21 (*Fgf21*) gene expression was significantly upregulated in HFD-fed *Atg7* knockout mice [109].

### 3.2. Autophagic Repressor in MAFLD

Autophagic repressors related to MAFLD include lipid receptor CD36, melatonin, leucine aminopeptidase 3 (LAP3), and SARS-CoV-2 Spike protein (Figure 1 and Table 1).

#### 3.2.1. Lipid Receptor CD36

Fatty acid transport proteins (FATPs) such as caveolin 1, a plasma membrane fatty acid binding protein (FABPpm), and the fatty acid translocase CD36 are implicated in cellular uptake of FFAs [110]. Bleriot, C. et al. showed that CD36 expression in the KC2 subpopulation was upregulated in HFD-induced obese mice. The transcriptomic profile of *Cd36* knockdown in mice liver showed that genes involved in lipid metabolism are significantly altered [25].

CD36 has been reported to inhibit lipophagy through an AMPK-dependent pathway [111]. HFD treatment for 14 weeks significantly increased the expression level of CD36 in male C57BL/6J mice and decreased the expression of the autophagosome marker LC3II in the liver, resulting in hepatic steatosis.

One study investigated the effect of CD36 on the development of NAFLD in both *Cd36* knockout and *Cd36* reconstituted mice. *Cd36* knockout resulted in an increase in LC3II in mouse liver, and reconstitution of *Cd36* restored the hepatic autophagy activity. Cell model experiments showed that overexpression of *CD36* in human hepatocyte HepG2 or Huh7 cell lines resulted in the accumulation of lipid droplets, whereas knockdown of *CD36* resulted in lipophagy and increased β-oxidation [111].

Li, Y. et al. subsequently analyzed the molecular mechanisms by which CD36 inhibits hepatic autophagy and found that HFD treatment increased the phosphorylation of AMPK in *Cd36* knockout mice. In HepG2 cells, *CD36* knockdown increased ULK1 expression and phosphorylation at Ser555. In Huh7 cells, *CD36* knockdown induced the expression of Beclin1 and phosphorylation at Ser93 [111].

#### 3.2.2. Leucine Aminopeptidase 3 (LAP3)

A study in 2019 comparing the plasma proteomic profiles of 48 patients with and without cirrhosis or NAFLD showed that the expression levels of six proteins were significantly altered. This study compared the plasma proteome profiles of HFD-induced mild fatty liver (HFD fed for 1–2 months) and severe fatty liver (HFD fed for more than 6 months) mice and showed that 40 plasma proteins increased and 31 proteins decreased in the severe fatty liver group. Of the 40 increased plasma proteins, 3 aminopeptidases (DPP4, LAP3 and ENPEP) were identified [112]. Of these, DPP4 is known to promote inflammation and insulin resistance in VAT [113].

Feng, L. et al. studied the role of LAP3 in the regulation of autophagy in HFD-fed E3 rats. They found that HFD treatment increased LAP3 expression in serum and liver of E3 rats with a NASH phenotype. CHO treatment increased LAP3 expression and inhibited the autophagy marker LC3 II/I ratio in L02 cells. In contrast, *LAP3* knockdown increased the LC3 II/I ratio in L02 cells. Treatment with the LAP3 inhibitor, bestatin, significantly increased the LC3 II/I ratio and the number of autophagic vesicles in L02 cells. One human study reported that the plasma level of LAP3 in patients with NAFLD was increased and positively correlated with fasting blood glucose and TG levels [114].

#### 3.2.3. SARS-CoV-2 Spike Protein

SARS-CoV-2 causes coronavirus disease 19 (COVID-19), a global public health problem since December 2019. Previous studies showed that lipid raft is essential for the replication of SARS-CoV-2 [115]. Decreased total CHO and low-density lipoprotein have been reported to be associated with the disease severity in COVID-19 patients [116,117,118]. This can lead to severe acute respiratory syndrome and death [119].

Recently, Shirazi Tehrani, A. et al. found that, the SARS-CoV-2 caused hepatocyte death and increased the expression of several autophagy markers (ATG5, LC3) and apoptosis markers (Bax, caspase 3) in the liver of postmortem cases [120]. Nguyen, V. et al. over-expressed the SARS-CoV-2 Spike protein in human embryonic kidney 293 cells (HEK293 cells), resulting in a marked increase in lipid deposition in the cell membrane and decreased expression of autophagy markers LC3 I and LC3 II [121].

**Table 1 ijms-23-10055-t001:** Summary of the effects of various autophagy modulators.

Factors	Effects	Types	Models	Mechanism of Action	Refs.
TH	induce	protein	C	Increase mitochondrial lipid oxidation and reduce inflammatory response in cultured HepG2-TRb cells	[50,51]
			M	Decrease serum CHO and ALT levels, hepatic steatosis, inflammation and fibrosis in NASH mice; increase hepatic mitochondrial content and function, hepatic autophagy to enhance b-oxidation of FFAs in NASH mice	
			H	Decreased hepatic steatosis in patients with NAFLD after treatment with low-dose levothyroxine or TRb-selective agonist, resmetirom (MGL-3196)	
irisin	induce	protein	C	Reduce the PA-dependent increasing of TGF-b1 and type I collagen expression and proinflammatory cytokines production	[55,60]
			M	Reduce the HFD-induced body weight, liver weight, AST and ALT levels, macrophage infiltration, proinflammatory cytokines production and NAFLD activity score; directly binding with MD2 to compete the binding of MD2 with TLR4	
			R	Restore the increased apoptotic cells, liver necrosis area and p62 expression level, and the decreased LC3B expression level in aged rats	
H_2_S	induce	gas	C	Decrease the lipid level and the apoptosis percentage for OA-treated liver cells and increase the expression levels of Beclin-1 and the LC3 II/I ratio	[69,70]
			M	Reduce the HFD-induced body weight, liver weight, white fat and brown fat weight, liver TC, TG, AST, ALT, lipid level, TNF-α, IL-1β, and IL-6 levels; inhibit the ROS/PI3K/AKT/mTOR signaling pathway in liver	
			CK	Decrease the ROS/PI3K/AKT/mTOR signaling pathway and increase the expression levels of Beclin1, ATG5 and LC3 II/I ratio	
melatonin	induce	protein	C	Decrease a-SMA, type 1 collagen, MMP9, TIMP1 and TGF-b expression levels in LX-2 cells and isolated primary HSCs	[80,81,82]
			M	Decrease serum AST, ALT, liver MDA, histopathological score and expression levels of cleaved caspase-3; increase LC3 II/I ratio and Beclin-1 expression levels	
DA-1241	induce	compound	M	Decrease serum CHO and liver TG in HFD mice	[83,84]
			H	Improve efficacy in the early clinical development for T2D treatment	
VMP1	induce	protein	M	*Vmp1* deletion in mouse liver impaired mitochondrial b-oxidation and caused the accumulation of neutral lipids in liver, decreased TG levels in serum and the secretion rate for TG; decreased VMP1 expression level in HFD-induced NASH mice	[87,88]
			Z	*vmp1* gene deficiency causes lipoprotein accumulation in the liver and intestine of zebrafish	
			H	Decreased hepatic VMP1 expression level in patients with NASH	
Nrf2	induce	protein	M	Nrf2 activator induced the Nrf2/HO-1 dependent pathway to inhibit cleaved caspase-3 and Bcl-2 expression, induce LC3B expression in I/R injury mouse model	[92,93,94]
			R	Nrf2 activator induced Nrf2/HO-1 axis and autophagy, decreasing inflammation, oxidative stress in I/R injury rat models	
SGLT-2i	induce	compound	C	Dapagliflozin reduced the lipid droplets contents in HepG2 and L02 cells	[96,97,99]
			M	Empagliflozin decreased fat mass, body weight, plasma TG and FFA levels, fasting blood glucose levels, NLRP3 inflammasome activity and induced HO-1-adiponectin dependent signaling pathway to prevent obesity	
			R	Dapagliflozin activated AMPK and inhibited mTOR in ZDF rat	
			H	SGLT-2i improved plasma liver enzymes, liver steatosis and NAFLD fibrosis score in T2DM and NAFLD patients	
*IRGM*	induce	gene	C	*IRGM* knockdown in HepG2 and PLC/PRF/5 cells decreased pULK1, Beclin-1 and LC3 II/I ratio and increased lipid droplet	[40,106]
			H	Obese children with variant *IRGM* rs10065172 genotype have higher risk of MAFLD	
*ATG7*	induce	gene	C	*ATG7* knockdown in hepatocytes increased the lipid droplet amount; *ATG7* V471A mutation significantly increased the expression levels of p62, and decreased LC3 II/I ratio in HepG2 cells	[107,108,109]
			M	HFD-fed *Atg7* knockout mice induced the myokines *Fgf21* gene expression	
			H	*ATG7* rs143545741 variants in severe NAFLD patients	
CD36	repress	protein	C	*CD36* overexpression in HepG2 or Huh7 cells increased lipid droplets accumulation whereas *CD36* knockdown induced lipophagy and fatty acid b-oxidation	[111]
			M	HFD increased hepatic CD36 expression and decreased LC3II expression in C57BL/6J mice; *Cd36* knockout increased LC3II in mouse liver and reconstruction of *Cd36* restored the autophagy inhibition	
LAP3	repress	protein	C	CHO treatment increased LAP3 expression levels and decreased LC3 II/I ratio in L02 cells	[114]
			R	HFD increased LAP3 expression level in the E3 rats with NASH; *LAP3* knockdown increased LC3 II/I ratio	
			H	Elevated plasm LAP3 levels in patients with NAFLD; positively correlated with fasting blood glucose and TG levels	
SARS-CoV-2 S	repress	protein	C	Increased lipid deposition in cell membrane, decreased expression levels of LC3I and II in SARS-CoV-2 Spike protein over-expressed HEK293 cells	[120,121]
			H	SARS-CoV-2 caused hepatocyte death and increased the expression levels of ATG5, LC3, Bax and caspase 3 in the liver of postmortem cases	

C: cell; M: mouse; R: rat; CK: chicken; H: human; Z: zebrafish. ALT: alanine aminotransferase; a-SMA: alpha-smooth muscle actin; AST: aspartate aminotransferase; *ATG*: autophagy-related gene; Bcl-2: B-cell lymphoma-2; CHO: cholesterol; FFAs: free fatty acids; HFD: high-fat diet; HO-1: heme oxygenase-1; HSCs: hepatic stellate cells; IL: interleukin; I/R: ischemia/reperfusion; IRGM: immunity-related GTPase M; LAP3: leucine aminopeptidase 3; LC3B: microtubule-associated protein 1 light chain-3B; MD2: myeloid differentiation factor 2; MMP: matrix metalloproteinase; mTOR: mammalian target of rapamycin; NAFLD: nonalcoholic fatty liver disease; ASH: nonalcoholic steatohepatitis; Nrf2: nuclear factor erythroid 2-related factor 2; OA: oleic acid; PA: palmitic acid; PI3K: phosphoinositide 3-kinases; pULK: phospho Unc-51 like autophagy activating kinase; Refs: references; ROS: reactive oxygen species; SARS-CoV-2 S: severe acute respiratory syndrome coronavirus 2 spike protein; SGLT-2i: sodium-glucose co-transporter type-2 inhibitors; T2D: type 2 diabetes; TC: total cholesterol; TG: triglyceride; TGF-b: transforming growth factor-beta; TH: thyroid hormone; TLR4: toll-like receptor 4; TNF-a: tumor necrosis factor alpha; TIMP: tissue inhibitors of matrix metalloproteases; TRb: thyroid hormone receptor; VMP: vacuole membrane protein.

### 3.3. Application of Autophagy-Modulating Agents in Human Diseases

Dysfunction and dysregulation of autophagy are associated with many human diseases such as metabolic syndrome, lung disease, kidney disease, infectious disease, cardiovascular disease, liver disease, neurodegenerative disease, and cancers. Drugs that modulate autophagic activity have the potential to treat these diseases [39,41,122,123,124]. Autophagy dysfunction increases the accumulation of abnormal proteins in cells and induces oxidative stress and ER stress, leading to genomic instability and tumor progression [41]. Takamura, A. et al. used systemic mosaic deletion of *Atg*5 and liver-specific *Atg*7 mice to develop multiple liver tumors. In contrast, p62 deletion partially restored the development of the tumor in liver-specific *Atg*7-deficient mice [125].

Several FDA-approved drugs are repurposed for autophagy regulation in cancer therapy [126]. Chloroquine (CQ) and hydroxychloroquine (HCQ) act as autophagy inhibitors to disrupt autophagosome-lysosome fusion, while rapamycin, metformin, pevonedistat and lithium chloride act by activating the AMPK pathway or increasing mTOR or glycogen synthase kinase 3 (GSK-3) in different cancer cells [41,127]. Among them, HCQ is currently in phase II/III trials, and CQ and pevonedistat are in phase III clinical trials [127].

Xu, X. et al. used a high-throughput screening method to investigate the effect of metformin in a mouse model of Alzheimer’s disease (AD). Metformin can activate the lysosome-dependent selective degradation pathway and chaperone-mediated autophagy (CMA) [128]. AD is the most common neurodegenerative disease leading to dementia, which is characterized by hyperphosphorylated tau pathology and Aβ peptide oligomerization [129]. Activation of CMA by the metformin-TAK1-IKKα/β Hsc70 signaling pathway reduced the level of accumulated Aβ plaques in the brain of AD mice, indicating that CMA activation to degrade Aβ may be a potential therapeutic strategy for AD [128]. Additionally, metformin has been shown to inhibit the development and recurrence of certain cancers [130].

## 4. Conclusions

The pathogenesis of MAFLD is complex and involves lipid accumulation, insulin resistance, inflammation, and fibrosis in the liver. During the development of MAFLD, hepatocytes, KCs, HSCs, and adipose tissue interact with each other. Chronic high-fat and high-sugar diets lead to excess FFAs and trigger CD36 transport to the hepatocyte membrane to take up FFAs into cells and inhibit the autophagy process. Lipid accumulation in hepatocytes leads to lipotoxicity, oxidative stress, and ER stress, and activates the ROS/PI3K/AKT/mTOR signaling pathway and triggers apoptosis in damaged hepatocytes. FFAs also activate M2 polarization in KCs to release TGF-β, in turn transforming HSCs to produce α-SMA, collagen type 1, and MMP9, leading to extracellular matrix accumulation and liver fibrosis.

Recent studies on the molecular mechanism of autophagy regulation in MAFLD have demonstrated that TH, irisin, H_2_S, melatonin, DA-1241, VMP1, Nrf2, SGLT-2i, *IRGM*, and *ATG7* have positive regulatory effects, while CD36, LAP3, and SARS-CoV-2 spike protein act as negative regulators. We performed an online protein–protein interaction network functional enrichment analysis at https://string-db.org/ using STRING (accessed on 26 July 2022) and found that ATG7, Beclin-1, mTOR, caspase 3, Nrf2, and p62 play central roles in protein interaction networks, whereas CD36, LAP3, irisin and melatonin receptor type 1B do not appear to interact with other proteins (Figure 2). In addition to high interactions between autophagy factors (ATG7 and MAP1LC3B, score 0.999) or apoptotic factors (AKT1 and MTOR, score 0.999), factors involved in autophagy and apoptosis are highly intersected with each other (AKT1 and BECN1, score 0.977). The scores of STRING interactions are listed in Appendix A.

This review article summarizes current knowledge and new advances related to the role of autophagy in MAFLD. Autophagy modulates major pathological changes, including hepatic lipid metabolism, inflammation, and fibrosis, suggesting the potential of modulating autophagy for the treatment of MAFLD. To date, although much has been learned about hepatic autophagy in the pathogenesis of MAFLD, there are still gaps in translating these molecular mechanisms into practical clinical applications. So far, there are no FDA approved drugs for MAFLD. In the future, by analyzing complex interactions between various molecular pathways, we will be better able to identify more effective therapeutic targets for developing drugs to treat MAFLD.

## Figures and Tables

**Figure 1 ijms-23-10055-f001:**
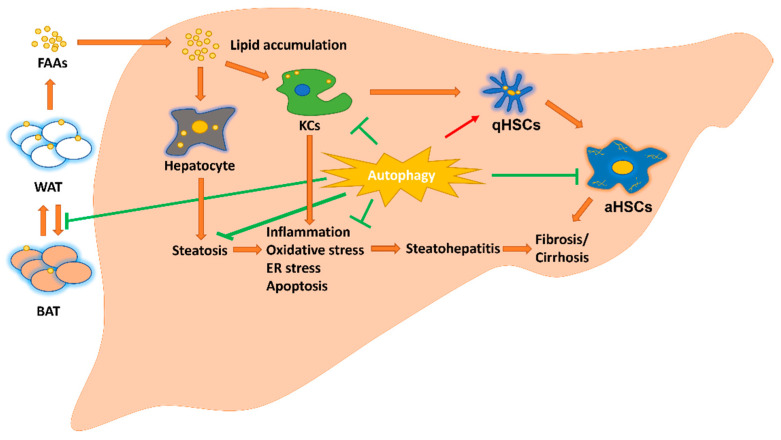
Regulation of autophagy on fatty liver pathogenic pathways. Autophagy inhibits hepatic steatosis, inflammation, oxidative stress, ER stress, apoptosis, and M1 KCs polarization. Autophagy induces M2 KCs. Autophagy may play a dual role in HSC activation. aHSCs: activated hepatic stellate cells; BAT: brown adipose tissue; ER: endoplasmic reticulum; FFAs: free fatty acid; KCs: Kupffer cells; qHSCs: quiescent hepatic stellate cells; WAT: white adipose tissue.

**Figure 2 ijms-23-10055-f002:**
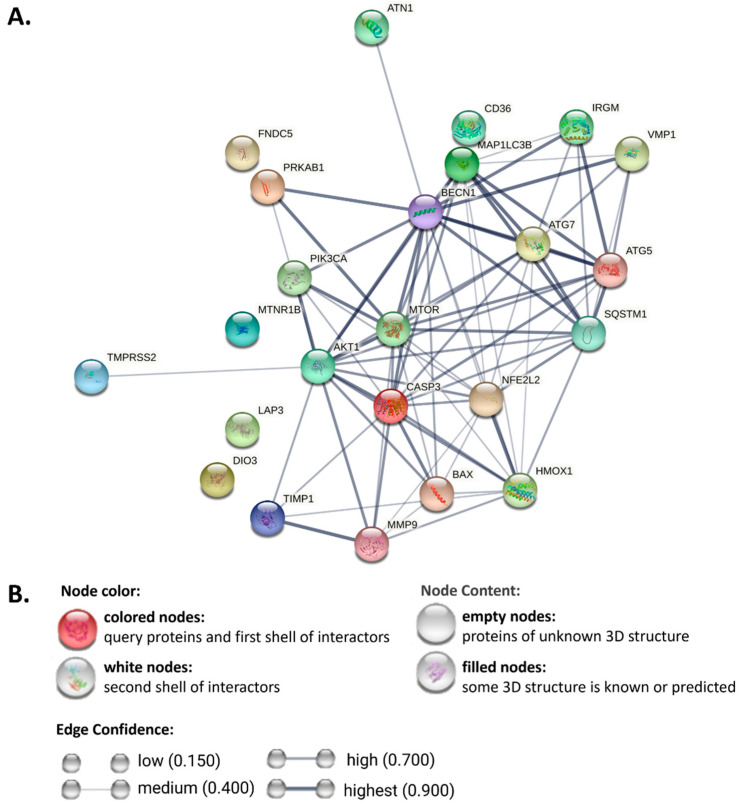
Functional enrichment analysis of protein–protein interaction networks. (**A**) Autophagy-related markers, proteins, and newly discovered autophagy regulators were selected to predict their interaction network via STRING. (**B**) The indicating colors and content of the nodes and the confidence of edges. ATN1: Atrophin-1; CASP3: caspase-3; DIO3: thyroid hormones; FNDC5: Irisin; HMOX1: HO-1; MTNR1B: melatonin receptor type 1B; NFE2L2: Nfr2; PIK3CA: PI3K; PRKAB1: AMPK; SQSTM1: p62; MAP1LC3B: LC3B; TMPRSS2: SARS-CoV-2 S protein.

## Data Availability

Source data are provided in this paper and are available from the corresponding author upon reasonable request.

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
