# Peer review of "Autophagy Dysregulation in Metabolic Associated Fatty Liver Disease: A New Therapeutic Target"

_ijms, 2022, doi:10.3390/ijms231710055_

Round 1

Reviewer 1 Report

I would like to thank to editors for giving me the opportunity to review the manuscript entitled “Autophagy dysregulation in metabolic associated fatty liver disease: a new therapeutic target".

The topic is relevant and well written; however, I think, there are some concerns that need to be addressed. Specific comments are as follows:

1.     Introduction

-       Line 49: References are missing. In addition, I recommend to describe to additional description about the reason why autophagy is important.

2.     Part 2-5: why are they included? If you want to explain the role of each cell or tissue in MAFLD, I recommend organizing them into one section and then into subgroups.

3.     Minors

A.     Please check the abbreviation: Atg or ATG

B.      Line 483: 6.1 conclusion -> 6.4. Conclusion or 7. Conclusion   

-       Although obesity and metabolic syndrome are independent risk factors for NAFLD, there are also many NAFLD patients with normal body mass index, that is, non-obese patients. Therefore, hepatic steatosis and any of overweight/obesity, presence of type 2 diabetes mellitus, or evidence of metabolic dysregulation is called MAFLD. Although an international experts suggest the change from NAFLD to MAFLD,

Author Response

  1. Introduction - Line 49: References are missing.

Response: Thanks for your kindly remind, we have added Ref. 14 [Singh R, Kaushik S, Wang Y, Xiang Y, Novak I, Komatsu M, et al. Autophagy regulates lipid metabolism. Nature 2009;458:1131-1135].

  1. In addition, I recommend to describe to additional description about the reason why autophagy is important.

Response: In this revised manuscript, we have added a description to highlight the importance of autophagy, as follows: “non-parenchymal cells of the liver, including KCs and HSCs also can utilize autophagy to maintain their homeostasis or function, thereby affecting pro-inflammatory and fibrotic responses in MAFLD progression” (page 2, line 58-60).

  1. Part 2-5: why are they included? If you want to explain the role of each cell or tissue in MAFLD, I recommend organizing them into one section and then into subgroups.

Response: In addition to hepatocytes, non-parenchymal cells of the liver, including KCs and HSCs, can also utilize autophagy to maintain their homeostasis or function, thereby affecting pro-inflammatory and fibrotic responses in MAFLD progression. Therefore, we thought it would be good to report the effects of these cells or tissues. In this revised manuscript, we have reorganized them as follows according to your suggestion:

Section 2: Hepatic parenchymal and non-parenchymal cells and adipose tissue in MAFLD (page 2, line 66)

2.1. Hepatocyte in MAFLD (page 2, line 69)

2.2. Macrophage in MAFLD (page 3, line 92)

2.3. Hepatic stellate cells (HSCs) in MAFLD (page 3, line 123)

2.4. Adipose tissue in MAFLD (page 4, line 142)

  1. Please check the abbreviation: Atg or ATG

Response: We modified the gene abbreviations according to the Guidelines for Formatting Gene and Protein Names (https://www.biosciencewriters.com/Guidelines-for-Formatting-Gene-and-Protein-Names.aspx) as follows:

       Gene name in human and chicken: ATG

       Gene name in mouse or rat: Atg

       Gene name in fish: atg

  1. Line 483: 6.1 conclusion -> 6.4. Conclusion or 7. Conclusion   

Response: In this revised manuscript, we have reorganized it into Section 4 (page 16, line 569).

  1. Although obesity and metabolic syndrome are independent risk factors for NAFLD, there are also many NAFLD patients with normal body mass index, that is, non-obese patients. Therefore, hepatic steatosis and any of overweight/obesity, presence of type 2 diabetes mellitus, or evidence of metabolic dysregulation is called MAFLD. Although an international experts suggest the change from NAFLD to MAFLD,

Response: Thanks for your recommendation. We have revised this sentence as follows: “MAFLD was previously known as non-alcoholic fatty liver disease (NAFLD), defined as the excessive deposition of fat in the liver in the absence of heavy alcohol consumption. Since fatty liver associated with metabolic dysfunction is common, "MAFLD" has recently recognized as a more appropriate overarching term.” in the Introduction section. (page 1, line 37 ~ page 2, line 39)

Reviewer 2 Report

Dear authors,

Manuscript ijms-1866624 is a Review manuscript entiteled "Autophagy dysregulation in metabolic associated fatty liver disease: a new therapeutic target" and authored by Chun-Liang Chen , Yu-Cheng Lin. The manuscript targets a hot topic that is potentially of high interest for the journal readers. The work seems to be accurately conducted and the gathered data covers the field of study. Unfortunately few issues needs authors attention before suggesting the manuscript for publication:

1. The abstract: please change this sentence "The effect of autophagy-modulating drugs and proteins on hepatic lipid metabolism may be a new therapeutic target for MAFLD." to a sentence that highlights main findings of the manuscript.

2. In the results section another figure gathering all informations about the involvement of hepatocytes, macrophage, hepatic stellate cells, adipose tissue and autophagy in MAFLD.2.

3. Figure 2 is non readable please improve the lisibility and resolution of this figure please.

4. the conclusion section have to be strengthen to feflect all the aspects covered by the review and mainly to highlight main findings,current gaps and  future directions in the field.

I will be waiting to read an improved version of this review that addresses all these issues and that I can recommand for publication

Best regards

Author Response

  1. The abstract: please change this sentence "The effect of autophagy-modulating drugs and proteins on hepatic lipid metabolism may be a new therapeutic target for MAFLD." to a sentence that highlights main findings of the manuscript.

Response: Thanks for your recommendation, we have revised this sentence as follows: “This review summarizes recent advances in the role of autophagy in MAFLD. Autophagy modulates major pathological changes, including hepatic lipid metabolism, inflammation, and fibrosis, suggesting the potential of modulating autophagy for the treatment of MAFLD.” in the Abstract section. (page 1, line 23-26).

  1. In the results section another figure gathering all information about the involvement of hepatocytes, macrophage, hepatic stellate cells, adipose tissue and autophagy in MAFLD.

Response: Thanks to your recommendation, we have modified Figure 1 to collect all information on the involvement of hepatocytes, macrophages, hepatic stellate cells, adipose tissue, and autophagy in MAFLD. (page 5, line 195).

  1. Figure 2 is non readable please improve the visibility and resolution of this figure please.

Response: We have modified Figure 2 to improve visibility and resolution. (page 17, line 593).

  1. the conclusion section have to be strengthen to reflect all the aspects covered by the review and mainly to highlight main findings, current gaps and future directions in the field.

Response: Thanks for your recommendation. In this revised manuscript, we have modified the Conclusions section as follows: “This review article summarizes current knowledge and new advances related to the role of autophagy in MAFLD. Autophagy modulates major pathological changes, including hepatic lipid metabolism, inflammation, and fibrosis, suggesting the potential of modulating autophagy for the treatment of MAFLD. To date, although much has been learned about hepatic autophagy in the pathogenesis of MAFLD, there are still gaps in translating these molecular mechanisms into practical clinical applications. So far, there are no FDA approved drugs for MAFLD. In the future, by analyzing complex interactions between various molecular pathways, we will be better able to identify more effective therapeutic targets for developing drugs to treat MAFLD.” (page 18, line 601 ~ 609)

Round 2

Reviewer 2 Report

Dear Authors,

I can now recommend your manuscript for publication.

Best regards